# Multilingual Lottery Tickets to Pretrain Language Models

**Jaeseong Lee and Seung-won Hwang**[*]
Computer Science and Engineering, Seoul National University
jslee@ldi.snu.ac.kr, seungwonh@snu.ac.kr

## Abstract

The curse of multilinguality in training multilingual pretrained language models (mPLMs) refers to the negative interference between languages, especially when the capacity is limited. While increasing the capacity may appear intuitive for overcoming this curse, it negatively affects both training and inference costs. Our distinction is pursuing the competing goals of reducing negative interference, while keeping the capacity per each language more or less the same. Specifically, we first scale the model to reduce interference, then search for a per-language subnetwork, or a lottery ticket, with comparable performance to the full model. According to lottery ticket hypothesis, this scale-then-find-ticket approach alleviates interfering signals as in the scaled model, but redistributes parameters to keep the parameters reduced. Finally, to avoid the cost of multiple retraining for searching multilingual tickets, we explore zero-shot neural architecture search (NAS) methods. We investigate the most appropriate zero-shot NAS method to find multilingual tickets. Our proposed *multilingual tickets* reduce the inference cost of models for each languages, while boosting the performances. The ticket search cost is negligible and tickets found qualitatively preserve linguistic similarity. Our code is publicly available.

## 1 Introduction

Multilingual pretrained language models (mPLMs) (Devlin et al., 2019; Conneau et al., 2020) have become the de-facto standard for multilingual tasks. These models, by mapping multiple languages through shared parameters (Figure 1a), may benefit from a positive transfer between languages (Pires et al., 2019; Conneau and Lample, 2019).

However, when more languages are covered, or parameters are limited, negative interference between languages (Wang et al., 2020b) has been observed, known as *curse of multilinguality* (Conneau et al., 2020).

For the problem of signals from languages interfering with each other, a naïve solution would be to increase the capacity per each language (Conneau et al., 2020; Pfeiffer et al., 2022). For example, Figure 1b illustrates adding additional parameters per each language to the shared model. This design intuitively improves the average performance of each language and allows the gradient conflict to be alleviated, but with the cost of enlarging per-language parameter size, which affects both the training and inference costs of the model.

Our distinction is keeping the per-language capacity similar to reach the same goal. To achieve multiple competing goals –improving the performance and mitigating the gradient conflict, without increasing per-language capacity– we invite the lottery ticket hypothesis (Frankle and Carbin, 2019). Lottery ticket hypothesis claims that dense models contain subnetworks –called "winning tickets"– whose performance is at least similar to the full model. This hypothesis is empirically shown to be true for popular language model architectures (Prasanna et al., 2020; Chen et al., 2020a; Zheng et al., 2022).

Our key idea is to search tickets per each language, i.e., *multilingual tickets*, that can achieve all of the competing goals, by scaling the model and then searching for such tickets. The parameters of each language will be redistributed in the scaled model to keep the per-language capacity unaffected by scaling. To illustrate, in Figure 1c, each of $l_1$ and $l_2$ maintains the same capacity before (Figure 1a) and after scaling (Figure 1c) as 9 parameters. This will i) improve the performance, since the performance from tickets will be similar to the scaled model, by the lottery ticket hypothesis. Moreover, disseminating the parameters will also ii) mitigate the negative interference.

---

[*]Corresponding author

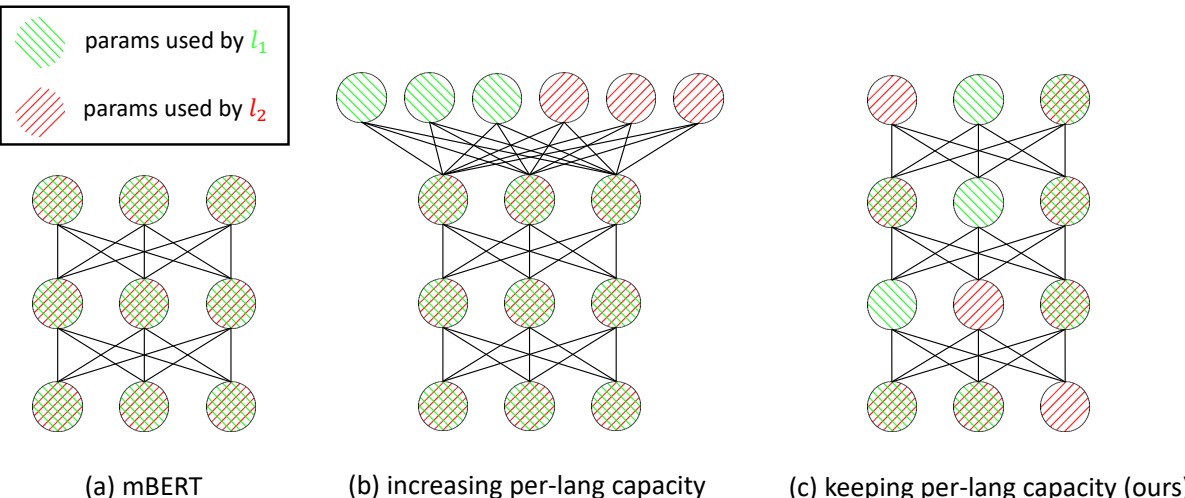

Figure 1: Comparison between various multilingual pretraining methods. (a): The naïve multilingual pretraining, where all the languages share all parameters. However it suffers negative interference between languages. (b): Increasing per-language capacity. (c): Keeping per-language capacity, while mitigating the interference between languages.

However, finding multilingual tickets in a scaled model incurs a prohibitive cost –even to find a single ticket, Frankle and Carbin (2019) train model multiple times. To overcome, we interpret this problem as neural architecture search (NAS) (Zoph and Le, 2017; Liu et al., 2019) per each language. Particularly, we explore recently emerging *zero-shot NAS* (Abdelfattah et al., 2021; Javaheripi et al., 2022; Shu et al., 2022), which aims to search architecture without training cost. We discriminate the most appropriate zero-shot NAS method to remove the burden of finding multilingual tickets.

Finally, to verify our claims, we devise a metric to measure the negative interference during multilingual pretraining. Our measurements indicate that *multilingual tickets* decrease negative interference. Our experiments show that multilingual tickets we found increase task performance while maintaining the capacity as expected. In our qualitative analysis, the locality of subnetworks preserves the linguistic similarity within the same family as well.

Our contributions can be summarized as follows:

- We propose a novel method to alleviate negative interference during multilingual pretraining: Searching *multilingual tickets*.

- We explore the most appropriate zero-shot NAS method to remove the cost of finding multilingual tickets.

- Experiments show that multilingual tickets do alleviate negative interference, increasing the task performance while keeping the capacity and computational complexity.

- Our code is publicly available. [1]

## 2 Preliminaries

### 2.1 Negative Interference in mPLM

Pretraining an mPLM utilizing corpora of multiple $N$ languages aims to leverage a positive cross-lingual transfer (Pires et al., 2019; Devlin et al., 2019; Conneau and Lample, 2019). However, Wang et al. (2020b) unveil a negative interference in mPLMs.

Specifically, they measure the interference from bilingual pretraining ($N = 2$), using cosine similarity between gradients (Yu et al., 2020), originally devised to measure the interference from multitask training. First, the total loss from multitask training $\mathcal{L}(\theta)$ can be denoted as a sum of each task loss $\mathcal{L}_i(\theta)$. Then the total gradient can be decomposed as follows:

$$\nabla \mathcal{L}(\theta) = \nabla \sum_i \mathcal{L}_i(\theta) = \sum_i g_i(\theta) \qquad (1)$$

where $g_i(\theta) = \nabla \mathcal{L}_i(\theta)$. They consider two tasks $i$ and $j$ interfere with each other if the cosine similarity between $g_i$ and $g_j$ is low. Wang et al. (2020b) use this metric to measure the interference in bilingual pretraining as follows:

$$gc(\theta) = \cos(g_1(\theta), g_2(\theta)) \qquad (2)$$

[1]https://github.com/thnkinbtfly/mtickets

where $g_1$ and $g_2$ are gradients from each language.

In this measure. the inference increases as $gc(\theta)$ becomes lower, which is maximized when $gc(\theta)$ is $-1$, indicating $g_1(\theta)$ and $g_2(\theta)$ are in opposite directions, canceling the updates for each other. Wang et al. (2020b) reveal that $gc(\theta)$ is lower in bilingual pretraining ($N = 2$) compared with monolingual pretraining ($N = 1$), implying negative interference happens during multilingual pretraining.

Our strategy is finding a lottery ticket for scaling up and reducing interference, shown in the following subsection.

## 2.2 Lottery Ticket Hypothesis

Lottery ticket hypothesis (Frankle and Carbin, 2019) states that every dense model contains some subnetwork whose performance is at least similar to the dense model. Formally, given the initial parameter of the dense model $\theta$, let $p(\theta)$ be the performance of the network. The Lottery ticket hypothesis is denoted as follows:

$$\exists m \in \{0, 1\}^{\dim \theta} \text{ s.t. } p(m \odot \theta) \geq p(\theta) \quad (3)$$

where $m \odot \theta$ denotes the pruned subnetwork, and $\dim \theta$ denotes the dimensionality of $\theta$. They named $m$ as "winning ticket"[2] of the given dense network.

Frankle and Carbin (2019) find the ticket by iterative train-and-prune. In each stage, they train the model starting from the subnetwork from the previous stage, and prune some parameters with the least magnitudes. They repeat this expensive process for multiple stages to get the final ticket $m$, known as iterative magnitude pruning.

Our key idea is finding tickets per each language, i.e., finding subnetwork $m_i \odot \theta$ for every language $l_i \in \{l_1, \cdots, l_N\}$. However, iterative magnitude pruning $N \approx 100$ languages in our target problem of multilingual pretraining incurs a prohibitive cost. Thus we formulate our target search as a zero-cost Neural Architecture Search (NAS) per each language, removing the cost of searching the architectures.

## 2.3 (Zero-shot) Neural Architecture Search

NAS (Zoph and Le, 2017; Zoph et al., 2018) aims to automatically search for superior architectures in the given search space (Elsken et al., 2019). While the seminal works, NASNet (Zoph et al., 2018) or AmoebaNet (Real et al., 2019), outperformed the

---

[2]For simplicity, we will abbreviate as "ticket".

manually-designed neural architectures, they were extremely computationally expensive –for example, AmoebaNet needed 7 days on 450 enterprise GPUs.

Therefore, reducing the search cost attracted keen research interests (Pham et al., 2018; Liu et al., 2019). Recently, zero-shot NAS (Abdelfattah et al., 2021; Mellor et al., 2021; Javaheripi et al., 2022) emerged, whose goal is to make the search cost almost negligible. We study how to leverage zero-shot NAS for our purpose of finding subnetworks per each language.

## 3 Proposed Method: Multilingual Lottery Tickets with Zero-Shot NAS

This section presents our method, scaling the model and then finding tickets $m_i$ per each language $l_i$ (§3.1), leveraging zero-shot NAS techniques (§3.2).

### 3.1 Scale-then-Search Multilingual Tickets

**Lower Interference, Higher Performance**  Allowing more space for each language to operate without interfering with each other has been shown to be beneficial in previous studies (Conneau et al., 2020; Pfeiffer et al., 2022). However, such a change would increase the total parameter size, which in turn increases both training and inference costs.

Our distinction is first to scale the baseline model, specifically by increasing the number of layers, from the initial model $\eta_0$ (3 in Figure 1a) to $\eta_s$ (4 in Figure 1c). We then redistribute the per-language parameters, by finding subnetworks maintaining the initial per-language parameter size.

However, would scaling the model from $\theta$ to $\theta'$ then finding the lottery ticket $m'$ from $\theta'$ perform better than $p(\theta)$? To answer this question, we reinterpret Equation 3. Since scaling (He et al., 2016; **?**; OpenAI, 2023) is a trustworthy method to improve performance, or, $p(\theta') > p(\theta)$, it has the effect of raising a lower bound performance of $p(m' \odot \theta')$.

In conclusion, once we successfully identify the per-language tickets from the scaled model, not only will the interference be alleviated, but the performance will also be enhanced. To ensure similar per-language capacity and computational complexity during the search procedure, it is crucial to carefully design the search space to apply neural architecture search (NAS) techniques.

**NAS Search Space**  Our goal in finding multilingual tickets is to keep computational complexity

and capacity, rather unaffected by scaling.

First, regarding computational complexity for matrix multiplication, we mask rows or columns from key and query matrices $K, Q$, respectively, for pruning subnetwork. When masking the entire row of $K$ (and a column $Q$) reduces the cost of matrix multiplication, we constrain masking rules to favor those with lower multiplication costs. Similarly, when we mask the rows of value matrix $V$ and columns of the following linear layer $W_0$ or, for the rows of $W_1$ and columns of $W_2$. The total masked capacity is constrained to be similar to the baseline model.

Formally, given input $h_i$, we mask the layers to get the output $h_o$ as follows:[3]

$$h_1 = \text{smax}(\frac{(\mu_q \cdot Qh_i)(\mu_q \cdot Kh_i)^T}{\sqrt{d}})(\mu_v \cdot Vh_i)$$
$$h_2 = (\mu_v \cdot W_0^T)^T h_1 + b_0$$
$$h_3 = LN_0(h_0 + h_1)$$
$$h_4 = (\mu_w \cdot W_2^T)^T \phi(\mu_w \cdot W_1 h_3 + \mu_w \cdot b_1) + b_2$$
$$h_o = LN_1(h_2 + h_4)$$

where $\cdot$ represents a scalar product along rows, smax denotes the softmax function, and $\phi$ is the activation function.

Second, regarding controlling capacity, we let the NAS algorithm search through the various candidate tickets of $m = \mu_q; \mu_v; \mu_w$, where ; is the concatenation. To maintain capacity and computational complexity, we set $r = |m|_0 / \dim(m) \approx \eta_0/\eta_s$, where $||_0$ denotes the number of non-zero components.

## 3.2 Zero-Shot NAS

**Choosing Zero-Shot NAS**   On the defined search space, we will search for tickets while minimizing the search cost. For this goal, we need to choose a specific zero-shot NAS algorithm. To narrow it down, we describe the characteristics of the zero-shot NAS method we need.

(a) **Input-adaptive**: Our goal is adapting subnetwork $m_i \odot \theta$ to languages. However, recent zero-shot NAS methods (Tanaka et al., 2020; Javaheripi et al., 2022; Zhou et al., 2022; Shu et al., 2022; Sun et al., 2022), such as Synflow or TF-TAS, aims to find a task-specific structure that remains invariant for inputs. For

---

[3]For simplicity, we describe with the transformer layer with a single-head attention. We similarly generalize to the transformer layers with multi-head attention.

input-invariance, they use the input filled with 1s, which is counter-intuitive to our purpose of finding different structures per different language inputs. We thus resort to input-adaptive zero-shot NAS methods.

(b) **Transformer-friendly**: Our search space is based on transformers (Vaswani et al., 2017). However, some NAS methods (Mellor et al., 2021; Lin et al., 2021a) rely on particular attributes of CNNs, since the NAS methods mainly developed from searching CNN architectures. For example, JACOV_COV (Mellor et al., 2021) assumes the network uses ReLU (Fukushima, 1969), which is typically true for CNNs. However NLP models based on transformers architecture (Devlin et al., 2019; Brown et al., 2020; OpenAI, 2023) mostly use GeLU (Hendrycks and Gimpel, 2016) instead of it. Such approaches are not practical for our goal.

To this end, we choose SNIP (Lee et al., 2019) as the most appropriate zero-shot NAS method for searching *multilingual tickets*. SNIP calculates the score of the subnetwork as follows:

$$S(\theta) = \sum_k S(\theta_k) = \sum_k \left| \frac{\partial \mathcal{L}}{\partial \theta_k} \odot \theta_k \right| \quad (4)$$

Note that this equation is the same as *importance* (Molchanov et al., 2019), when we consider measuring the scores of $m$ only (Michel et al., 2019). Importantly, this score relies on the given inputs, which satisfies (a). Moreover, SNIP (or, importance) has use cases in transformer architectures (Michel et al., 2019; Prasanna et al., 2020), which supports (b).

**Searching Multilingual Tickets with SNIP**
With the NAS search space defined, and the zero-shot NAS chosen, we will search tickets per each language $l_i$, i.e., determine the subnetwork $m_i \odot \theta$ with the highest SNIP score (Eq. 4). Formally, we need to maximize as follows:

$$\max_{m_i} S(m_i \odot \theta) = \max_{m_i} \sum_k \left| \frac{\partial \mathcal{L}}{\partial m_{i,k}} m_{i,k} \right| \quad (5)$$

Since $S(m_{i,k})$ is always non-negative, and $m_{i,k} \in \{0, 1\}$, maximizing the $\sum_k S(\theta_k)$ is setting $m_{i,k} = 1$ when $S(m_{i,k})$ is within the top $r\%$

of the SNIP values.[4] Therefore, once we collect the gradients $\frac{\partial \mathcal{L}}{\partial m_{i,k}}$ with the initial weight $\theta$ and given input data from $l_i$, we can easily decide the ticket $m_i$.

### 3.3 On Stably Measuring Negative Interference

Though a commonly used measurement of negative interference is cosine similarity between gradients (Eq. 2), denoted as $gc$ (orange) in Figure 2, its measurement is highly variant, or the metric change is significant over training steps.

Such a variance makes comparing the reported performance at the given step highly unreliable. We thus propose to compare a cumulative metric: We evaluate the cosine similarity between the cumulated parameter updates by each language. Fortunately in our case, since each batch consists of a single language, we can easily decompose the total updates from languages. Formally, suppose the language $l^{(t)}$ is used in each step $t$. Then we define our metric to evaluate negative interference as follows:

$$u_i(\theta_T) = \sum_t^T \mathbb{1}(l^{(t)} = l_i)\delta_t \quad (6)$$

$$uc_{i,j}(\theta_T) = \cos(u_i(\theta_T), u_j(\theta_T)) \quad (7)$$

where $\mathbb{1}(l^{(t)} = l_i)$ is 1 if and only if $l^{(t)} = l_i$, and $\delta_t$ is the parameter update from step $t$. We regard negative interference between language $l_i$ and $l_j$ is larger as $uc_{i,j}(\theta_T)$ is smaller.[5]

Our proposed metric compares an accumulated effect of the influences, which is more tolerant to steps: In Figure 2, we show the relative difference of cumulative metric $uc$ (red) is much stabler than $gc$ (orange). Thus, we will use metric $uc$ in the following experimental section.

## 4 Experiments

In this section, we tackle the following research questions:

• RQ1: Do multilingual tickets improve performance?

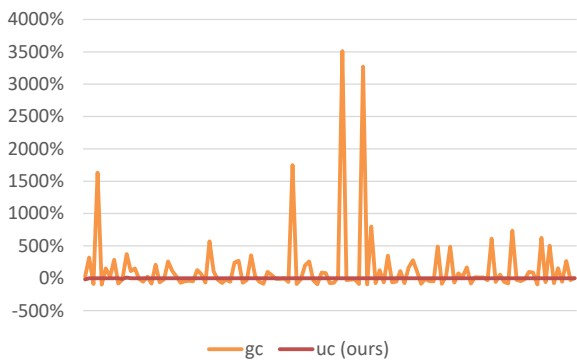

Figure 2: Relative difference of metrics for negative interference per every 10 steps in mBERT pretraining. Our proposed metric (red) is much stabler than cosine similarity between gradients (orange).

• RQ2: Are the searched multilingual tickets better than random tickets?

• RQ3: Do multilingual tickets mitigate the negative interference?

• RQ4: Is our method better than input-invariant NAS method?

• RQ5: How much more computations would be needed to match our improvement with naïve mBERT?

### 4.1 Experimental Settings

**Unlabeled Datasets and Languages for Pretraining** We utilize Wikipedia dumps of the same languages mBERT (Devlin et al., 2019) used. We extract Wikipedia articles using WIKIEXTRACTOR. To split the sentences, we use STANZA (Qi et al., 2020) for Thai (th), and TRANKIT (Nguyen et al., 2021) for Hebrew (he), Armenian (hy), and Arabic (ar). We also use Arabic version of TRANKIT for azb, ka, my, ne, new, pnb. Otherwise, we split the sentences using MOSES (Koehn et al., 2007).

**Task Datasets and Languages for Evaluation** We focus on evaluating in-language performance using XTREME benchmarks (Hu et al., 2020; Ruder et al., 2021). Since we focus on in-language performance, we deal with the NER and POS tasks, which are the only tasks available for tens of languages in the benchmarks. Moreover, we restrict the languages with a sufficient amount of train and test data, for reliable evaluation.[6] These results in evaluating 42 languages over 14 language families

---

[4]We normalize the SNIP score by layers, similar to the normalization of importance score by Michel et al. (2019). We choose the top $r\%$ values after collecting SNIP values over all languages.

[5]Note that the positive transfer would be larger as $uc_{i,j}(\theta_T)$ becomes larger, following the convention of $gc$ (Wang et al., 2020b).

[6]We omit the language from the given task, if the train or test data size is not larger than 100.

| | en | de | fr | nl | ru | es | it | pl | ja | vi | uk | ar | pt | fa | id |
|---|---|---|---|---|---|---|---|---|---|---|---|---|---|---|---|
| mBERT | 80.39 | 86.30 | 87.69 | 88.61 | 85.35 | 89.31 | 88.46 | 88.12 | 61.45 | 87.68 | 89.99 | 84.34 | 89.35 | 90.43 | 90.08 |
| random tickets ($\eta_s = 14$) | 80.35 | 86.31 | 88.11 | 88.86 | 85.26 | 89.30 | 88.50 | 88.35 | 61.19 | 88.04 | 89.79 | 84.59 | 89.22 | 90.14 | 90.23 |
| random tickets ($\eta_s = 16$) | 80.22 | 86.04 | 87.80 | 88.81 | 85.21 | 89.03 | 88.07 | 88.05 | 61.03 | 87.78 | 89.84 | 84.07 | 89.34 | 90.54 | 90.10 |
| multilingual tickets ($\eta_s = 14$) | 80.56 | 86.27 | 87.90 | 88.91 | 85.17 | 89.33 | 88.79 | 88.37 | 61.54 | 88.15 | 89.93 | 84.53 | 89.22 | 90.26 | 90.20 |
| multilingual tickets ($\eta_s = 16$) | 81.05 | 86.75 | 88.51 | 89.30 | 85.92 | 89.70 | 89.01 | 88.62 | 61.68 | 88.24 | 90.59 | 85.10 | 89.89 | 90.81 | 90.54 |

| | ko | fi | tr | hu | ro | eu | ms | he | bg | kk | et | el | lt | az | ur |
|---|---|---|---|---|---|---|---|---|---|---|---|---|---|---|---|
| mBERT | 82.89 | 88.16 | 89.98 | 89.53 | 91.48 | 87.99 | 90.61 | 79.03 | 89.95 | 81.23 | 88.29 | 87.04 | 85.97 | 84.22 | 91.88 |
| random tickets ($\eta_s = 14$) | 82.68 | 88.31 | 89.87 | 89.91 | 91.54 | 88.05 | 91.84 | 78.88 | 89.73 | 80.86 | 88.37 | 86.85 | 85.75 | 83.28 | 91.77 |
| random tickets ($\eta_s = 16$) | 82.80 | 88.01 | 89.85 | 89.81 | 91.63 | 88.05 | 90.22 | 78.51 | 89.95 | 81.89 | 88.28 | 87.04 | 85.84 | 84.18 | 90.86 |
| multilingual tickets ($\eta_s = 14$) | 83.35 | 88.61 | 90.01 | 90.12 | 91.97 | 88.31 | 90.20 | 79.57 | 89.89 | 82.69 | 88.69 | 87.05 | 86.39 | 84.40 | 91.51 |
| multilingual tickets ($\eta_s = 16$) | 83.66 | 88.90 | 90.40 | 90.46 | 91.89 | 88.63 | 92.04 | 80.13 | 90.77 | 82.06 | 89.09 | 87.77 | 86.99 | 85.86 | 91.61 |

| | ka | hi | th | ta | bn | af | mr | ml | te | sw | tl | zh | avg | FLOPS(M) |
|---|---|---|---|---|---|---|---|---|---|---|---|---|---|---|
| mBERT | 80.69 | 76.37 | 50.78 | 79.06 | 91.17 | 84.96 | 78.19 | 76.48 | 72.81 | 85.32 | 92.59 | 72.73 | 84.21 | 341.7 |
| random tickets ($\eta_s = 14$) | 79.64 | 76.10 | 49.42 | 77.55 | 89.77 | 83.67 | 75.35 | 76.02 | 69.76 | 86.51 | 92.50 | 72.32 | 83.92 | 338.6 |
| random tickets ($\eta_s = 16$) | 80.45 | 75.72 | 48.20 | 78.45 | 90.34 | 83.90 | 76.59 | 76.00 | 70.70 | 86.57 | 91.96 | 73.06 | 83.92 | 341.3 |
| multilingual tickets ($\eta_s = 14$) | 81.05 | 76.36 | 47.18 | 78.00 | 90.33 | 85.67 | 78.81 | 76.82 | 73.09 | 86.02 | 93.16 | 72.60 | 84.31 | 335.5 |
| multilingual tickets ($\eta_s = 16$) | 81.72 | 77.06 | 52.34 | 79.38 | 90.99 | 86.34 | 79.89 | 76.70 | 72.73 | 86.90 | 92.93 | 73.25 | **84.91** | 337.7 |

Table 1: F1 scores on NER task over 42 languages. We average the score over 5 runs.

and 1 isolate: af, ar, az, bg, bn, de, el, en, es, et, eu, fa, fi, fr, he, hi, hu, id, it, ja, ka, kk, ko, lt, ml, mr, ms, nl, pl, pt, ro, ru, sw, ta, te, th, tl, tr, uk, ur, vi, zh.

**Implementation Details** The baseline dense model follows the BERT architecture (Devlin et al., 2019). We use $\eta_0 = 12$ layers, with 768 hidden units where the intermediate linear layer $W_1$ expands the dimension to 3072. We scale the model so that $\eta_s$ becomes 14 or 16. When searching for multilingual tickets, we accumulate gradients from 2.5M tokens to calculate the SNIP values (Eq. 4). This takes only about 2-3 minutes on 1 RTX 3090 per language.

To pretrain the model, we follow the default setting described in Devlin et al. (2019). We over-sample low-resource languages with an exponential smoothing factor of 0.7. We use a learning rate of 1e-4, batch size of 256, and update for 1M steps. We use a sequence length of 128 for 90% of the updates and 512 for the last 10%. Pretraining is conducted on TPUv3-8.

Our evaluation settings largely follow the XTREME benchmarks (Hu et al., 2020; Ruder et al., 2021). We fine-tune the pretrained models with a batch size of 32 and a learning rate of 2e-5. We generally fine-tune them for two epochs. But for NER, since the training dataset for some languages is scarce, we ensure to update parameters for at least 2500 iterations. We run 5 times per each language, and report the average score over all languages. Fine-tuning is conducted on RTX 3090.

**Comparisons** We compare the following models:

- mBERT: Naïve multilingual pretraining with the baseline dense model architecture.

- random tickets: Multilingual pretraining utilizing the randomly selected tickets on the defined search space.

- multilingual tickets: Multilingual pretraining with the multilingual tickets found on the defined search space.

### 4.2 Experimental Results

**RQ1: Effectiveness of Multilingual Tickets** The results presented in Tables 1 and 2 demonstrate the significant computational efficiency of our proposed multilingual ticket variants compared to other approaches. For example, multilingual tickets ($\eta_s = 16$) outperforms mBERT, while requiring fewer FLOPs for inference.

**RQ2: Effectiveness of Chosen Zero-Shot NAS** To demonstrate the efficacy of our chosen zero-shot NAS approach, we compare it with randomly selected tickets from the same search space. Table 1 and 2 highlight the substantial performance improvement achieved by our cost-free search method; For instance, random tickets ($\eta_s = 16$) even suffer performance degradation when compared to mBERT, while our multilingual tickets outperform it.

**RQ3: Alleviated Interference by multilingual tickets** To confirm that the discovered multilingual tickets mitigate negative interference, we compare the average $uc$ (Eq. 7) values. In a controlled

| | en | de | fr | nl | ru | es | it | pl | ja | vi | uk | ar | pt | fa | id | ko | fi |
|---|---|---|---|---|---|---|---|---|---|---|---|---|---|---|---|---|---|
| mBERT | 94.86 | 97.08 | 95.65 | 96.25 | 97.58 | 95.33 | 95.66 | 97.77 | 87.39 | 84.67 | 95.40 | 82.70 | 93.18 | 94.63 | 77.78 | 76.71 | 91.93 |
| random tickets ($\eta_s = 14$) | 94.85 | 97.08 | 95.64 | 95.94 | 97.60 | 95.26 | 95.69 | 97.76 | 87.33 | 84.94 | 95.34 | 82.40 | 93.20 | 94.34 | 77.66 | 76.62 | 91.96 |
| random tickets ($\eta_s = 16$) | 94.94 | 97.06 | 95.60 | 96.18 | 97.62 | 95.31 | 95.69 | 97.75 | 86.93 | 85.25 | 95.48 | 82.62 | 93.20 | 94.61 | 77.67 | 76.57 | 92.02 |
| multilingual tickets ($\eta_s = 14$) | 94.90 | 97.08 | 95.69 | 96.26 | 97.64 | 95.29 | 95.75 | 97.87 | 87.31 | 85.44 | 95.74 | 83.14 | 93.24 | 94.79 | 77.78 | 76.77 | 92.26 |
| multilingual tickets ($\eta_s = 16$) | 94.91 | 97.09 | 95.67 | 96.43 | 97.73 | 95.32 | 95.78 | 97.91 | 87.34 | 85.24 | 95.80 | 84.08 | 93.24 | 95.24 | 77.92 | 76.98 | 92.39 |

| | tr | hu | ro | eu | he | bg | et | el | lt | ur | hi | ta | af | zh | avg | FLOPS(M) |
|---|---|---|---|---|---|---|---|---|---|---|---|---|---|---|---|---|
| mBERT | 75.86 | 94.10 | 96.35 | 89.55 | 88.73 | 97.86 | 94.62 | 97.19 | 89.57 | 91.15 | 90.73 | 83.40 | 96.70 | 87.82 | 91.23 | 341.7 |
| random tickets ($\eta_s = 14$) | 76.23 | 93.70 | 96.29 | 90.10 | 88.51 | 97.94 | 94.74 | 96.77 | 89.68 | 90.72 | 90.52 | 82.88 | 96.73 | 87.93 | 91.17 | 338.6 |
| random tickets ($\eta_s = 16$) | 76.10 | 94.20 | 96.40 | 90.20 | 88.82 | 97.90 | 94.79 | 97.05 | 90.05 | 90.80 | 90.56 | 82.40 | 96.70 | 88.04 | 91.24 | 341.3 |
| multilingual tickets ($\eta_s = 14$) | 76.42 | 94.01 | 96.42 | 90.41 | 88.79 | 98.10 | 94.98 | 97.01 | 90.04 | 90.78 | 90.92 | 82.91 | 96.89 | 88.26 | 91.38 | 335.5 |
| multilingual tickets ($\eta_s = 16$) | 76.69 | 94.50 | 96.48 | 90.58 | 89.66 | 98.05 | 95.05 | 97.27 | 90.70 | 91.09 | 91.00 | 82.83 | 96.97 | 88.28 | **91.55** | 337.7 |

Table 2: F1 scores on POS task over 31 languages. We average the score over 5 runs.

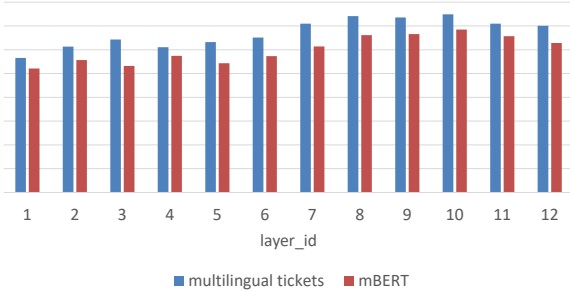

Figure 3: Multilingual tickets alleviates the negative interference, compared with mBERT. x-axis: layer id, y-axis: metric $uc$ (§3.3) indicating how much interference is mitigated.

| | input | NER | POS | FLOPS |
|---|---|---|---|---|
| magnitude pruning | invariant | 84.73 | 91.31 | 342 |
| SNIP (ours) | adaptive | **84.91** | **91.55** | **338** |

Table 3: Comparison between multilingual tickets with input-invariant zero-shot NAS (magnitude pruning) and input-adaptive zero-shot NAS (SNIP). We report the averaged NER and POS F1 score over all languages.

| | NER | POS | FLOPS |
|---|---|---|---|
| mBERT | 84.21 | 91.23 | 342 |
| mBERT scaled | 84.85 | 91.36 | 372 |
| multilingual tickets | **84.91** | **91.55** | **338** |

Table 4: Comparison between multilingual tickets and scaled mBERT. We report the averaged NER and POS F1 score over all languages.

experiment, we search for the multilingual tickets on the baseline model without scaling. With a setting of $r = 0.85$, we calculate the average $uc$ for each layer at 10K update steps, considering all languages.

As depicted in Figure 3, the $uc$ values of the multilingual tickets are higher compared to mBERT, indicating a reduction in negative interference.

**RQ4: Superiority of Input-Adaptive NAS** We emphasize that the selection of input-adaptive NAS is important. We establish another comparison of multilingual tickets built with a representative input-invariant zero-shot NAS method, magnitude pruning (Frankle et al., 2021). We set $\eta_s$ as 16 for both methods.

Table 3 shows that SNIP, the input-adaptive zero-shot NAS method, outperforms the magni-

tude pruning, the input-invariant zero-shot NAS method. This highlights that input-adaptive NAS is essential.

**RQ5: Computational Efficiency of Our Method** To estimate our improvement as computational cost, we compare with a scaled version of naïvely pretrained mBERT. We scale hidden units by 10%. Table 4 shows that even if we add 10% of FLOPs to the baseline, multilingual tickets outperforms it.

### 4.3 Analysis: Ticket Similarity and Language Relatedness

We further analyze the effectiveness of our multilingual tickets and provide deeper insights into their characteristics. Our hypothesis is that the multilingual tickets capture language relatedness, as related languages are less prone to negative interference (Wang et al., 2020b) and benefit more from the positive transfer (Pires et al., 2019; Khemchandani et al., 2021; Muller et al., 2021). The redistribution strategy of parameters (Figure 1c), if effective, should thus favor sharing parameters among related languages.

To investigate whether our multilingual tickets exhibit such characteristics, we project the found tickets for each language using UMAP (McInnes et al., 2018). For comparison, we also project randomly selected tickets. We identify language families using the Glottolog database (Hammarström et al., 2021) to list related languages.

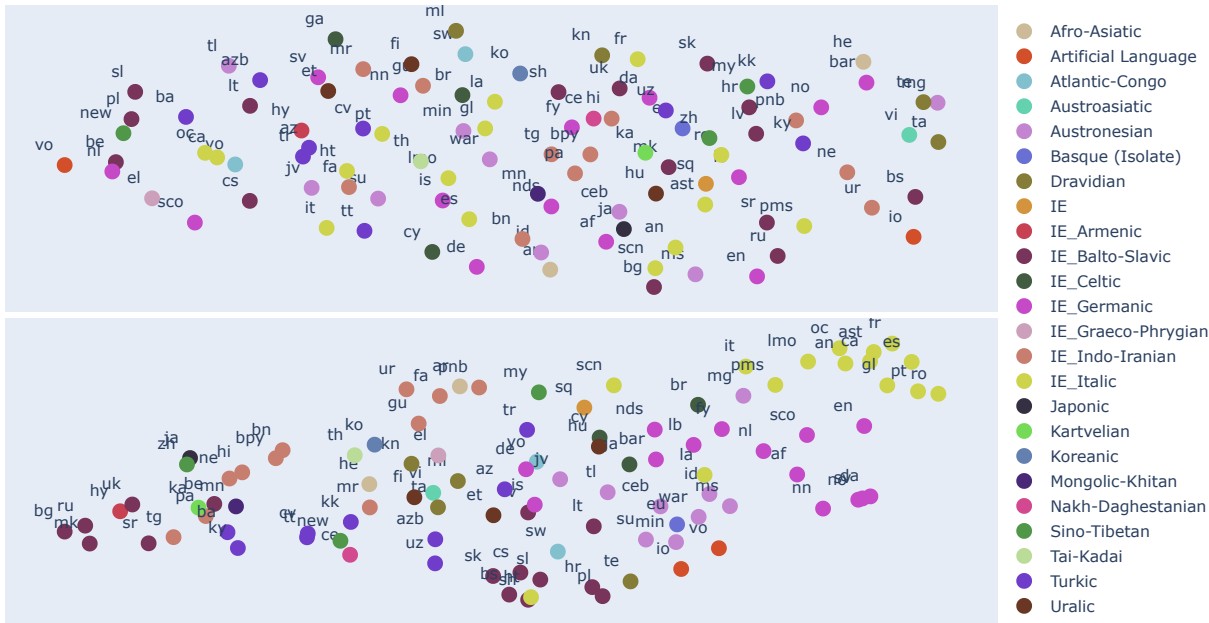

Figure 4: Random tickets (top) and the found multilingual tickets (bottom) projected via UMAP. Each color corresponds to a specific language family. Multilingual tickets from same language families are similar to each other.

Figure 4 illustrates the results, where our multilingual tickets (bottom) show similarity among languages from the same language family (dots with the same color). This is in stark contrast to the random tickets (top). For example, in our multilingual tickets, the tickets of the artificially constructed languages Volapük (vo) and Ido (io) are close to each other, while they are positioned at the left and right edges in the random tickets. Similarly, natural languages belonging to the Germanic language family exhibit similar characteristics.

One might question the Balto-Slavic languages as a counter-example, as they appear in two clusters at the bottom of Figure 4. However, with a more fine-grained taxonomy, we can further verify that the multilingual tickets correlate with linguistic genealogy. The Balto-Slavic languages on the left side belong to the East Slavic or Eastern South Slavic languages, while the languages on the right side belong to the West Slavic, Western South Slavic, and Baltic languages. One exception is Serbian (sr), which belongs to Western South Slavic but is closer to the left cluster. We hypothesize that since Serbian shares dialects, such as Torlak, with some Eastern South Slavic languages (Kortmann and van der Auwera, 2011), it is positioned closer to the left cluster rather than other Western South Slavic languages.

In summary, our multilingual tickets automati-cally learn language relatedness with only a few unlabeled language tokens and negligible computational cost. This finding greatly benefits multilingual pretraining.

## 5 Related Work

### 5.1 Subnetworks for mPLMs

Several previous works have explored the use of language-specific subnetworks in mPLM. For the translation task, Lin et al. (2021b) identify language-specific subnetwork based on the magnitude of neurons after some update, then further fine-tune it with the data of specific language pairs. Xie et al. (2021) investigate important neurons using importance score, then follow a similar approach. However, all of these assume that the negative interference incurred in the pretraining stage can be alleviated in a post-hoc manner, which is challenged by Pfeiffer et al. (2022). Our distinction is eradicating the interference from the pretraining stage, such that the post-hoc methods can be applied complementarily to our approach. The closest work to ours is S3Net (Lu et al., 2022), which utilizes language-specific subnetworks to pretrained mPLMs for automatic speech recognition (ASR). However it requires significant computational cost to find such subnetworks, whereas our method removes the need for expensive subnetwork search using zero-shot NAS.

## 5.2 NAS for Pretrained Language Models

After the success of automatically searched neural networks (Zoph et al., 2018; Real et al., 2019) outperforming the manually-designed architectures, researchers have also applied NAS for NLP models (So et al., 2019; Wang et al., 2020a; So et al., 2021; Gao et al., 2022; Javaheripi et al., 2022). While most of these works focus on optimizing models for a single language, DARTS-ASR (Chen et al., 2020b) leverages a successful NAS method, DARTS (Liu et al., 2019), to automatically search multilingual ASR model for four languages. Similarly, Tsai et al. (2020) perform NAS to find a shared architecture for multilingual corpora. In contrast, our approach leverages NAS techniques to address negative interference by searching for language-specific tickets rather than focusing on the shared model part, as overlooked in previous works. Moreover, to the best of our knowledge, we are the first to explore zero-shot NAS methods for mPLMs.

## 6 Conclusion

This paper studied the challenge by balancing the conflicting goals of reducing negative interference while maintaining a similar per-language capacity. We proposed a Scale-then-Search approach, of searching for per-language subnetworks, or lottery tickets, from a scaled model, that improves performance without increasing per-language capacity. We keep the cost of finding such tickets negligible, by exploring a zero-shot NAS method. Our results show that ours reduces negative interference as expected, and the tickets discovered qualitatively preserve linguistic relatedness.

## Limitation

**Generalization to Unseen Languages**  Our research primarily focuses on the effectiveness of multilingual tickets and their impact on reducing interference. However, the performance and generalizability of our approach to languages unseen during the pretraining stage, may be limited. Further investigation and adaptation of the method specifically for resource-poor settings are necessary.

**Fine-Grained Language Relatedness**  While we use language relatedness to qualitatively analyze multilingual tickets found, this notion may not capture fine-grained variations in language relatedness, which may require additional research to sophisticate qualitative analysis.

## Acknowledgements

This research was partially supported by the MSIT (Ministry of Science and ICT), Korea, under the ITRC (Information Technology Research Center) support program (IITP-2023-2020-0-01789) supervised by the IITP (Institute for Information & Communications Technology Planning & Evaluation). This research was also partially supported by a gift from Google. We would also like to thank Google's TPU Research Cloud (TRC) program for providing Cloud TPUs.

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
