# OpenReview forum: "Multilingual Lottery Tickets to Pretrain Language Models"
_EMNLP/2023/Conference — EMNLP 2023 Findings_

### Official Review · Reviewer_kJa9 · 2023-08-03

**Soundness:** 3

**Excitement:**

3: Ambivalent: It has merits (e.g., it reports state-of-the-art results, the idea is nice), but there are key weaknesses (e.g., it describes incremental work), and it can significantly benefit from another round of revision. However, I won't object to accepting it if my co-reviewers champion it.

**Missing References:**

Ansell, Alan, et al. "Composable sparse fine-tuning for cross-lingual transfer." arXiv preprint arXiv:2110.07560 (2021).


**Paper Topic And Main Contributions:**

This paper tries to address the curse of multi-linguality in multiLMs, while maitaining the same #parameters and #FLOPS as the baseline model. They do by first scaling up the language model (compared to the mBERT model), and then searching for language-specific lottery tickets of the scaled model. They constrain the final lottery tickets to have the same number of parameters as the baseline model, which is mBERT in this paper. They propose a computationally-cheap appraoch for finding lottery tickets based on zero-shot neural architecture search (NAS). They perform NAS by computing SNIP scores for language-specific inputs and then keeping the parameters with highest SNIP values. They show that the resulting lottery tickets perform better than both the baseline model (i.e., mBERT), as well as random tickets. (although the improvement is marginal).
They also propose a method for measuring gradient interference in the resulting model and show that the resulting tickets interfere less than the baseline model.
Finally they perform an analysis on the resulting lottery tickets and show that close languages (i.e., languages in the same language family) have more overlapping sub-networks.

**Questions For The Authors:**

- Is the proposed metric for gradient interference just a batched version of `gc` to reduce noise?
- Is the scaled-up model trained from scratch, or do you use the first 12 layers fixed from mBERT? Figure 1 might be a bit misleading if it's from scratch.
- Did you compare your NAS approach with something like (non-iterative) magnitude pruning? If so, how does it perform?



**Reasons To Accept:**

- A computationally-cheap appraoch for finding sub-networks could be interesting for the community, as e.g., methods based on iterative magnitude pruning, while having good performance, are quite expensive.
- The analysis on the overlap of similar languages' subnetworks is quite interesting and supports previous works that found language-neutral subnetworks for multiLMs.

**Reasons To Reject:**

- The idea of training large and then compressing is generally not novel. It has been investigated by [1], which they also specifically used cheap methods for doing the compression. Moreover, when scaling up the LM, it has to be pretrained from the beginning which is quite computationally-expensive. Is the cost of pre-training much lower than finding language-specific subnetworks with a relatively cheap approach?
- pruning with a ratio of 12/14 or even 12/16 can even be done with simple magnitude-pruning (without losing considerable performance), so we don't really know the significance of the proposed approach compared to trivial baseline methods. It would be nice if there was a comparison to some baseline methods, rather than only "random tickets".
- The improvement over the baseline method at the end is quite marginal, and the final tickets are also having the same #parameters as the original model, so the only real benefit would be a slight decrease in gradient interference (which doesn't really translate into better empirical performance for e.g. low/high-resource languages).


[1] Li, Zhuohan, et al. "Train big, then compress: Rethinking model size for efficient training and inference of transformers."

**Reproducibility:**

4: Could mostly reproduce the results, but there may be some variation because of sample variance or minor variations in their interpretation of the protocol or method.

**Reviewer Confidence:**

4: Quite sure. I tried to check the important points carefully. It's unlikely, though conceivable, that I missed something that should affect my ratings.

---

> ### Author Rebuttal · Authors · 2023-08-29
>
> We are glad that you find our work is computationally-cheap, and provides interesting analysis. We will reflect your suggestions (updating figure, adding reference) in the final version. We address your specific questions below.
>
> C1-i. “The idea of training large and then compressing is generally not novel (Zhuohan et al. 2020)” Our contribution is not following the old idea of training large and compressing (LeCun et al. 1989, Han et al. 2016). Our aim is reducing interference between languages, by providing tickets differently per language.
>
> C1-ii. “Is the cost of pre-training from the beginning much lower than finding language-specific subnetworks with a relatively cheap approach?” Comparing with the methods applied after pre-training challenged by previous work, for failing to remove the negative interference incurred in the pretraining stage (line 473).
> Our additional experiment shows that performing magnitude pruning applied after pretraining, though cheap, performs poorly (ex. F1 avg : NER -22, POS -18 from ours).
> Moreover, magnitude pruning with pretraining (and 10% further training compared to ours) is both more expensive and less effective (ex. F1 avg: NER - 2, POS -1 from ours).
> We thus focus on finding subnetworks before pre-training (We report magnitude pruning before pretrain in our answer below)
>
> C2. Q3. “More baselines, such as magnitude pruning, would be desirable” We reiterate that our aim is redistributing the parameters differently per language to avoid negative interference. Therefore, we stated in section 3.2 that generating the same ticket per language (as in magnitude-based pruning) is not desirable. As the table below reveals, ours **outperforms** magnitude pruning (Frankle et al. 2021). We highlight that POS performance of magnitude pruning is on par with mBERT, while ours improves with a notable amount.
>
> |                   | POS   | NER   | FLOPs(M) |
> | ----------------- | ----- | ----- | ------ |
> | mBERT             | 91.23 | 84.21 | 342    |
> | magnitude pruning | 91.31 | 84.73 | 342    |
> | ours              | 91.55 | 84.91 | 338    |
>
> C3. “The improvement over baseline is quite marginal” Our improvement, when translated into additional computational overhead baselines have to pay, is not marginal– For example. mBERT, even after trained with 10% more FLOPs than ours, performs far below than ours, in our additional evaluation reported below (limited by rebuttal period; We will report more extensive results in the revised draft).
>
> |              | POS   | FLOPs (M) |
> | ------------ | ----- | ------ |
> | mBERT        | 91.23 | 342    |
> | mBERT scaled | 91.36 | 372    |
> | ours         | 91.55 | 338    |
>
> Q1. “Is the proposed metric just a batched version of gc?” No. We compare the parameter updates, not the gradients. In this way, we can exactly compare the effect of each language to the optimization result.
>
> Q2. “Is the scaled-up model trained from scratch, or do you use the first 12 layers from the mBRET?” We trained from scratch.
>
> Y. LeCun, J. Denker, and S. Solla, “Optimal Brain Damage,” in NIPS 1989.
> S. Han, H. Mao, and W. J. Dally, “Deep Compression: Compressing Deep Neural Network with Pruning, Trained Quantization and Huffman Coding.,” in ICLR 2016.
> J. Frankle, G. K. Dziugaite, D. Roy, and M. Carbin, “Pruning neural networks at initialization: Why are we missing the mark?,” in ICLR 2021

---

### Official Review · Reviewer_spkU · 2023-08-05

**Soundness:** 5

**Excitement:**

5: Transformative: This paper is likely to change its subfield or computational linguistics broadly. It should be considered for a best paper award. This paper changes the current understanding of some phenomenon, shows a widely held practice to be erroneous in someway, enables a promising direction of research for a (broad or narrow) topic, or creates an exciting new technique.

**Paper Topic And Main Contributions:**

-

**Reasons To Accept:**

-

**Reasons To Reject:**

-

**Reproducibility:**

5: Could easily reproduce the results.

**Reviewer Confidence:**

1: Not my area, or paper was hard for me to understand. My evaluation is just an educated guess.

---

> ### Author Rebuttal · Authors · 2023-08-29
>
> Thank you for your positive evaluation. We will make sure to thoroughly revise to reflect all reviewer comments.

---

### Official Review · Reviewer_TbBM · 2023-08-05

**Soundness:** 3

**Excitement:**

4: Strong: This paper deepens the understanding of some phenomenon or lowers the barriers to an existing research direction.

**Paper Topic And Main Contributions:**

This paper studies the problem of multilingual curse in multilingual pretrained language models (mPLM), that is, the negative interference between different languages under limited capacity. The authors propose a novel approach to mitigate negative noise by searching for multilingual tickets while keeping the capabilities of each language similar. To reduce the cost of searching for multilingual tickets, the authors explore a zero-shot neural architecture search (NAS) approach. Experimental results demonstrate that the approach can mitigate negative perturbations and improve task performance while maintaining capacity and computational complexity.

**Reasons To Accept:**

1. A novel approach is proposed to address the multilingual curse in multilingual pre-trained language models, namely mitigating negative noise by searching for multilingual tickets.
2. Successfully exploited the zero-shot neural architecture search (NAS) method to reduce the cost of searching for multilingual tickets.
3. Experimental results show that the method can effectively mitigate negative interference and improve task performance while maintaining capacity and computational complexity.

**Reasons To Reject:**

1. The method in this paper may face challenges when dealing with more languages or more complex tasks, and further verification of its effectiveness in large-scale scenarios is required.
2. The experiments in this paper mainly focus on the effect of mitigating negative interference, but the impact on positive transfer has not been fully discussed.

**Reproducibility:**

3: Could reproduce the results with some difficulty. The settings of parameters are underspecified or subjectively determined; the training/evaluation data are not widely available.

**Reviewer Confidence:**

2: Willing to defend my evaluation, but it is fairly likely that I missed some details, didn't understand some central points, or can't be sure about the novelty of the work.

---

> ### Author Rebuttal · Authors · 2023-08-29
>
> We are encouraged that you find our paper to be novel, exciting and successful. We also point out your concerns are already addressed in the submitted draft, as below:
>
> C1. “The method in this paper may face challenges when dealing with more languages or tasks” Results we reported in the draft were **in the largest-scale setting (103 languages) reported in the literature** (i.e., most of existing works use less than 103), using the widely adopted XTREME-R benchmark for large-scale multilingual evaluation.
>
> C2. “The method mainly focuses on the effect of mitigating negative interference, but the positive transfer needs to be discussed” Positive transfer **is already discussed** in our manuscript. i) Figure 4 shows that our tickets preserve linguistic similarity, which is a critical signal for positive transfer, as suggested in previous works (line 416). ii) Our proposed metric ‘uc’ considers positive transfer also. High value of uc of ours (Figure 3) implies that positive transfer is large. uc measures both negative and positive transfer in one equation, rooted from the convention of previous works (Wang et al. 2020), and we will revise to emphasize this point in the camera ready.
>
> Z. Wang, Z. C. Lipton, and Y. Tsvetkov, “On Negative Interference in Multilingual Models: Findings and A Meta-Learning Treatment,” in 2020 EMNLP

---

### Meta-Review · Area_Chair_ZuGd · 2023-09-18

**Recommendation:** 3

**Metareview:**

The submission deals with language model pre-training.

The reviewers have praised the paper's approach to the task in hand as computationally efficient but they have also identified a number of shortcomings described in detail in the reviews.

---

### Decision · Program_Chairs · 2023-10-07

**Decision:**

Accept-Findings

**Comment:**

The submission deals with language model pre-training.

The reviewers have praised the paper's approach to the task in hand as computationally efficient but they have also identified a number of shortcomings described in detail in the reviews.